# Gradient-guided discrete walk-jump sampling for biological sequence generation

**Zarif Ikram**                                              *zarifikram003@gmail.com*
*CSE, BUET*
*National University of Singapore*

**Dianbo Liu**[*]                                            *dianbo@nus.edu.sg*
*National University of Singapore*

**M Saifur Rahman**[*]                                       *mrahman@cse.buet.ac.bd*
*CSE, BUET*

**Reviewed on OpenReview:** *https://openreview.net/forum?id=fFVuo4SPfT*

## Abstract

In this work, we propose gradient-guided discrete walk-jump sampling (gg-dWJS), a novel discrete sequence generation method for biological sequence optimization. Leveraging gradient guidance in the noisy manifold, we sample from the smoothed data manifold by applying discretized Markov chain Monte Carlo (MCMC) using a denoising model with the gradient-guidance from a discriminative model. This is followed by jumping to the discrete data manifold using a conditional one-step denoising. We showcase our method in two different modalities: discrete image and biological sequence involving antibody and peptide sequence generation tasks in the single objective and multi-objective setting. Through evaluation on these tasks, we show that our method generates high-quality samples that are well-optimized for specific tasks.

## 1 Introduction

Biological sequences like antibody sequences is critical for a broad range of problems involving diverse practical applications, from molecular imaging of cancer (Wu, 2014) to immunotherapy of cancer and viral diseases (Hudson & Souriau, 2003; Lu et al., 2020). The optimization is challenging: the protein sequence has an enormous state space with high-entropy variable regions. Experimental validation, on the other hand, is both time-consuming and expensive. Generally, the engineering process begins with *in vitro* experiments to determine aspects of interest such as humanization, specificity (Makowski et al., 2022), efficacy, and pharmacokinetic properties, followed by mutation to the samples for improved properties.

*Ab initio* sequence generation is a fascinating direction for generating novel biological sequences given prior samples. Using a generative model, these methods attempt to generate sequences that are similar to prior data. We can roughly divide these models into two groups: autoregressive models (Wang et al., 2022; Jain et al., 2022) and denoising models (Luo et al., 2022; Gruver et al., 2023). Albeit compelling, both come with their drawbacks. On one hand, autoregressive models suffer from error accumulation and even scalability in the case of reward distribution fitting (e.g., Jain et al. (2022)). On the other hand, denoising models require intricate noise scheduling, making the real discovery task difficult.

To combat the inefficiency of the autoregressive and denoising models, discrete walk-jump sampling (dWJS) (Frey et al., 2024) recently proposed sampling antibody sequences by walking on noisy manifolds, followed by denoising jumping to the discrete data manifold. While dWJS benefits from sampling from noisy manifolds for discrete sequence generation, its only objective is to sample from the data distribution, leading to

---

[*]These authors co-supervised the work.

sequences that are not necessarily optimized for a chosen attribute. However, in a real-world antibody optimization setting, we are interested in sequences that are not only antibody-like but also attribute-optimized. Instead of focusing on a single attribute, multi-objective optimization is important for biologicial seqeunce design. For example, therapeutic antibodies must meet numerous criteria at the same time, such as binding affinity, specificity, stability, and manufacturability. Traditional single-objective optimization can produce sequences that excel in one attribute but fail in others. By including multi-objective optimization in the generative process, we can balance competing objectives and build sequences that match all of the criteria. This method assures that the antibodies produced are not only effective, but also practical for real-world use. A simple but effective way of optimizing properties in a generative model is to train a discriminative model and use its gradient to guide the sampling process towards a high-fitness region. While this approach is simpler in continuous settings, enabling direct optimization in the antibody structure space, it still requires consideration of the amino acid sequence for actual synthesis, necessitating inverse-folding (Dauparas et al., 2022) of structures. However, the optimized structure does not guarantee a realizable sequence, and even if the sequence exists, there is no assurance of inverse-folding it, motivating the need for direct optimization in discrete sequence space. In a discrete setting such as antibody sequence generation, gradient guidance is difficult. The sampling step in dWJS, thankfully, takes place in the smoothed data manifold, which lets these discriminative models provide gradient guidance.

In this work, we present gradient-guided discrete walk-jump sampling (gg-dWJS), a novel approach for *ab initio* sequence generation, building on (Frey et al., 2024), as shown in Figure 1. To improve the likelihood of a sequence attribute, gg-dWJS learns a denoising and discriminative model on the noisy data manifold and then **walks** on the noisy data manifold using discretized Langevin MCMC with **gradient guidance**. Finally, using the model pair, our method performs one-step conditional denoising to **jump** back to the true data manifold. We present our approach in two modalities. First, we showcase our approach in discrete image domain in Section 5. Next, in Section 6, we detail our approach in antibody sequence generation with two examples: single objective optimization and multi objective optimization with preference conditioning. Finally, in Section 6.4, we showcase our method in a peptide generation task.

To summarize, the contributions of this work are:

- We propose gg-dWJS, a novel algorithm based on dWJS for target optimization using a noised discriminator model.

- Using preference conditioning, we formalize our method for use in a multi-objective optimization setting.

- We evaluate our method on discretized image, antibody sequence, and peptide generation tasks, showing that our method generates high-quality samples with optimized attributes.

## 2 Preliminary

### 2.1 Walk-jump sampling (WJS) with neural empirical Bayes (NEB)

By combining kernel density estimation and empirical Bayes, NEB (Saremi & Hyvärinen, 2019) provides a denoising method to recover $X$ for a smoothed random variable $Y = X + \mathcal{N}(0, \sigma^2 I_d)$ such that $X \rightharpoonup Y$ using the gradient log density of $Y$, where $X \rightharpoonup Y$ refers to the kernel density estimation smoothing. Formally, to retrieve $Y \rightharpoonup X$, given Y=y, we can get the least square estimator of $X = \hat{x}$ according to Robbins (1992); Miyasawa et al. (1961)–

$$\hat{x} = y + \sigma^2 \nabla \log p(y) \tag{1}$$

Where $\nabla \log p(y)$ is known as the *score function* (Hyvärinen & Dayan, 2005) which is the gradient of log density in the smoothed function $p(y) = \int p(y|x)p(x)dx$.

Overall, the above constitutes the WJS framework, where the walk part draws exact samples from the smoothed density by Langevin MCMC and the jump part approximates the least-squares estimator of X at an arbitrary time.

## 2.2 Discrete walk-jump sampling (dWJS)

A key property of the walk-jump sampling framework is that the least square estimation (jump) part is decoupled from the Langevin MCMC (walk) part. dWJS takes advantage of this decoupling to train an energy based model or a score based model on the smoothed distribution on the noised samples, sample noisy samples with Langevin MCMC, and return back to the denoised samples with a least square estimator. In our work, we restrict ourselves to score based model for both the sampler and the least square estimator.

Specfically, building on the NEB, dWJS learns the score function on the noisy data manifold parameterized as $g_\phi : \mathbb{R}^d \to \mathbb{R}^d$. Hence, equation 1 now looks like the following:

$$\hat{x} = y + \sigma^2 g_\phi(y) \tag{2}$$

We can learn the parameter $\phi$ by performing stochastic gradient descent on the following loss function to denoise $y$ in noisy data manifold to $x$ in discrete data manifold .

$$\mathcal{L}(\phi) = \mathbb{E}_{x \sim p(x), y \sim p(y|x)} ||x - \hat{x_\phi}(y)||^2 \tag{3}$$

To sample discrete under this formulation, we perform Langevin MCMC using $g_\phi$ on the noisy data manifold, returning to the true data manifold using equation 2.

## 2.3 Classifier guidance in generative models

To obtain a truncation-like in diffusion models, Dhariwal & Nichol (2021) modifies the diffusion score of a model with an auxilary gradient from the log likelihood of a classifier model. The modified score is then used during sampling from the diffusion model, which affects sampling by upweighting the data the classifier labels with high probability to the chosen label, providing a conditioning to the sampling process. We adapt this effect in our method by incorporating a discriminator model in the sampling process. Unlike the classfier guidance in diffusion models that uses during sampling, our method incorporates gradient guidance during the least square estimation (jump) too.

## 2.4 Multi-objective optimization

Multi-objective optimization (MOO) refers to discovering a set of candidates $x^* \in \mathcal{X}$ such that for $d$ different objectives $R(x) = [R_1(x), R_2(x), \ldots, R_d(x)]$, it maximizes them simultaneously, i.e.,

$$R(x^*) = \max_{x \in \mathcal{X}} R(x) = [\max_{x \in \mathcal{X}} R_1(x), \max_{x \in \mathcal{X}} R_2(x), \ldots, \max_{x \in \mathcal{X}} R_d(x)]$$

If the objectives are in tension with one another, there is no $x^*$ that simultaneously maximizes all objectives. This leads to the *Pareto optimality* in MOO, ensuring optimal trade-off among objectives. In this formulation, for a candidate $x_1$ to *dominate* $x_2$, written as $x_1 \succ x_2$,

$$x_1 \succ x_2 \iff R_i(x_1) \geq R_i(x_2) \forall i \in \{1 \ldots, d\} \land R_j(x_1) > R_j(x_2) \exists j \in \{1, \ldots, d\}.$$

Finally,

$$x^* \text{is } Pareto\text{-}optimal \iff x_i \nsucc x^* \forall i \in \{1, \ldots, d\}$$

*Pareto set* constitutes the Pareto-optimal candidates, and *Pareto front* defines their images in the objective space. A popular method to tackle MOO is *Scalarization*, which decomposes the MOO problem into a single-objective problem. In this method, we assign convex weight $w_i$ to each objective $R_i(x)$ such that $w_i > 0$ and $\sum_{i=1}^d w_i = 1$. Using the weights, we can derive a single-objective formulation, i.e., $\max_{x \in \mathcal{X}} R(x|w)$, where $R(x|w)$ is the *scalarization function*. A widely used scalarization function is *weighted-sum scalarization* (Ehrgott, 2005; Miettinen, 1999), where $R(x, w) = \sum_{i=1}^d w_i R_i(x)$. Using this formulation, we can represent the MOO problem as a family of subproblems, each defined by a different *preference vector $w$*. Under certain assumptions, the solutions to these problems are the Pareto optimal candidates for the preference vector w.

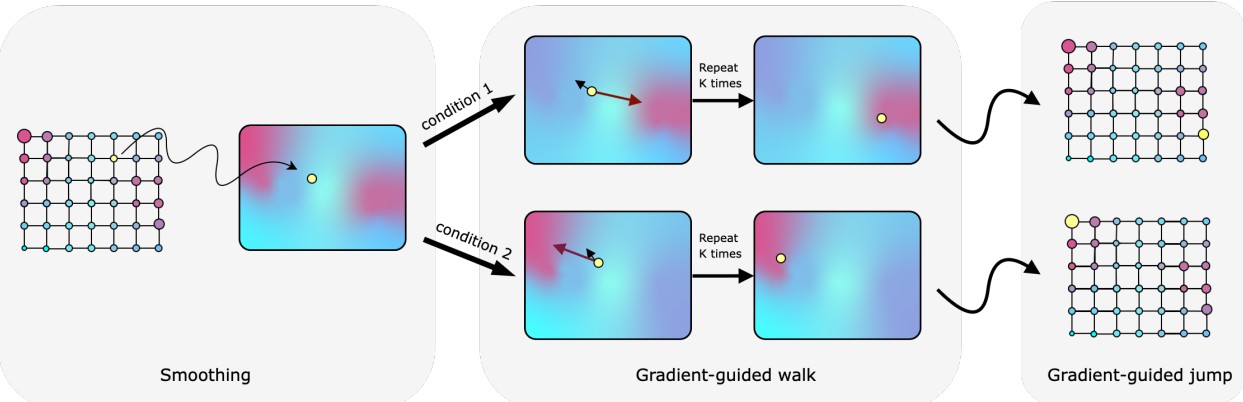

Figure 1: Gradient-guided discrete walk-jump sampling process. We begin the sampling process by smoothing some discrete seed. Next, we conduct the gradient-guided walk process by combining denoising gradient with the discriminator gradient. Finally, we perform gradient-guided jump to return to the discrete data manifold. Here, purple tint represents higher data density and larger circle represents higher data distribution.

## 3 Gradient-guided discrete walk-jump sampling

gg-dWJS augments dWJS by learning a discriminator model $P(C|Y)$ **(Step A)** on the noisy data $Y$ given the true data $X$ and its label $C$. Next, we train the score model $g_\phi$ on the noisy data $Y$ **(Step B)**. Later, we utilize the discriminator's gradient to guide the Langevin MCMC walk **(Step C)**. Finally, we jump from the noisy data manifold to the true discrete data manifold using the gradient guidance **(Step D)**. For the remainder of the paper, we use the following notations for the random variables: $X$ refers to the true data, $Y$ refers to the noisy data, and C refers to the class labels. The lowercase of of the said variables refers to the specific instances of them.

### 3.1 Step A: Learning a noisy discriminator model

Given the true data $X$ and its label $C$, we smooth $X$ by adding Gaussian noise, obtaining $Y = X + \mathcal{N}(0, \sigma^2 I_d)$. Next, we learn the discriminator model $\log P(C|Y)$ by parameterizing it with $f_\theta(Y)$. Note that $C$ can be both categorical and continuous. For example, for categorical values, we parameterize $f_\theta(y)$ to return logits over the labels given $y$, while for a specific attribute (e.g., Instability index), we simply learn the $f_\theta(y)$ over the attribute.

### 3.2 Step B: Learning a noisy score model

We learn the score model on the noisy data $Y$, parameterizing it with $g_\phi$. We learn the model by minimizing the loss $\mathcal{L}$ from equation 3.

### 3.3 Step C: Gradient guided walk

We start by sampling random discrete data $y_0 \sim \mathcal{U}_d$. Langevin MCMC (Sachs et al., 2017) uses the gradient of log density to guide the sampling process by providing information about the structure of the target distribution. In our method, we perform discretized Langevin MCMC on $y$ with gradient guidance from both score function $g_\phi(y)$ and discriminator $f_\theta(y)$. Formally, we use

$$g_\phi(y_k) + \lambda \nabla_Y f_\theta(y_k, c)$$

to perform MCMC for $K$ steps given $y_k$ and label $c$ at the $k_{\text{th}}$ step, where $\lambda$ is the relative gradient guidance strength with which we control the effect of gradient guidance.. Intuitively, one can see this as performing a gradient accent on the label $c$'s probability with respect to the input $y_k$ to find the local maxima while walking towards the smoothed data density with $g_\phi(y_k)$. We summarize the process in algorithm 1. For a

detailed explanation on the discretization of Langevin MCMC, we guide our readers to Sachs et al. (2017). Besides, we report all our hyperparameters in the Table 9.

---

**Algorithm 1** gradient-guided discrete walk-jump sampling

---

**Input:**

$c$: targetted label

$\delta$: step size

$u$: inverse mass

$\gamma$: friction

$\lambda$: relative gradient-guidance strength

$K$: number of steps taken

$g_\phi \approx \nabla \log p(y)$: learnt score function

$f_\theta \approx \log p(c|y)$: learnt discriminator model

$y_0 \sim \mathcal{N}(0, \sigma^2 I_d) + \mathcal{U}_d(0, 1)$

$v_0 \leftarrow 0$

**for** $k = 0, \ldots, K-1$;          `// gradient-guided walk`

 **do**

    $y_{k+1} \leftarrow y_k + \frac{\sigma}{2} v_k$

    $s_{k+1} \leftarrow g_\phi(y_{k+1})$ ;          `// score of` $y_{k+1}$

    $d_{k+1} \leftarrow \nabla f_\theta(y_{k+1}, c)$ ;      `// discriminator gradient guidance`

    $g_{k+1} \leftarrow s_{k+1} + \lambda d_{k+1}$

    $v_{k+1} \leftarrow v_k + \frac{u\delta}{2} g_{k+1}$

    $\epsilon \sim \mathcal{N}(0, I_d)$

    $v_{k+1} \leftarrow e^{-\gamma\delta} v_{k+1} + \frac{u\delta}{2} g_{k+1} + \sqrt{u(1 - e^{-2\gamma\delta})}\epsilon$

    $y_{k+1} \leftarrow y_{k+1} + \frac{\delta}{2} v_{k+1}$

$\hat{x}_K \leftarrow y_K + \sigma^2 g_\phi(y_K) + \sigma^2 \nabla f_\theta(y_K, c)$ ;      `// gradient-guided jump`

Perform `argmax` to receive the discrete sequence from $\hat{x}_K$

---

### 3.4 Step D: Gradient guided jump

After the $K$ steps of walking, we jump from the noisy data manifold to the clear discrete data using the gradient guidance and score model given smoothed data $y_k$ and label $c$. From NEB, given a smoothed input $y$, we can jump back to $\hat{x}$ using the equation 1. Without loss of generality, we can extend that, for a conditional variable $c$ and $y \sim p(y|c)$,

$$
\begin{aligned}
\hat{x} &= y + \sigma^2 \nabla_y \log p(y|c) \\
&= y + \sigma^2 \nabla_y \log p(c|y) + \sigma^2 \nabla_y \log p(y) - \sigma^2 \nabla_y \log p(c) \\
&= y + \sigma^2 \nabla_y \log p(c|y) + \sigma^2 \nabla_y \log p(y) \\
&\approx y + \sigma^2 \nabla_y f_\theta(y, c) + \sigma^2 g_\phi(y)
\end{aligned}
$$

Therefore, according to the above demonstration, we perform a one-step denoising to return to the true discrete data manifold using $f_\theta$ and $g_\phi$ from steps A & B. Specifically, the jump step approximates the true discrete point $\hat{x}$ by adding the gradient-guided adjustments to $y_K$ where $y_K$ is a point at any $K$ time in the walk process

$$\hat{x} = y_K + \sigma^2 \nabla_y f_\theta(y_K, c) + \sigma^2 g_\phi(y_K). \tag{4}$$

Here, $\sigma^2 \nabla_y f_\theta(y_K, c)$ represents gradient from the score model, helping align with the data distribution and $\sigma^2 g_\phi(y_K)$ provides attribute-specific guidance, pushing the output toward regions optimized for the desired attribute. This approach is useful because it allows to sample discrete points at any time of the walk process.

## 4 Multi-objective optimization with gg-dWJS

Given the added conditioning capability, we can now proceed to add the formulation for preference conditioning to our method. In the case of preference conditioning, since $\sum_{i=1}^{d} w_i = 1$, optimizing a single objective $R(x, w) = \sum_{i=1}^{d} w_i R_i(x)$ refers to a single point in the pareto front. This means that if we have a generative model that is controllable based on the preference vector $w = \{w_1, w_2, \ldots, w_d\}$, we can generate the solutions to the sub-problems, thereby constructing the pareto front.

A naive approach for controllable generation with preference conditioning is to simply train $d$ noisy objective function models $f_i(y, w) \approx w R_i(y)$ and use $d$ gradients to guide the generative process, i.e.,

$$g = g_\phi(y) + \nabla_y f_{1,\theta}(y, w_1) + \nabla_y f_{2,\theta}(y, w_2) + \cdots + \nabla_y f_{d,\theta}(y, w_d) \tag{5}$$

However, this approach requires us to train d discriminator models. Instead, we can approximate the combined discriminator and use its gradient in the following way:

$$
\begin{aligned}
g &= g_\phi(y) + \nabla_y f_{1,\theta}(y, w_1) + \nabla_y f_{2,\theta}(y, w_2) + \cdots + \nabla_y f_{d,\theta}(y, w_d) \\
&= g_\phi(y) + \nabla_y w_1 R_1(y) + \nabla_y w_2 R_2(y) + \cdots + \nabla_y w_d R_d(y) \\
&= g_\phi(y) + \nabla_y (w_1 R_1(y) + w_2 R_2(y) + \cdots + w_d R_d(y)) \\
&= g_\phi(y) + \nabla_y f_\theta(y, w)
\end{aligned}
$$

Here, $f_\theta(y, w) \approx w_1 R_1(y) + w_2 R_2(y) + \cdots + w_d R_d(y)$ is the preference-conditioned single-objective function. By learning this function approximation, we can have two benefits: (a) we need only one discriminator training, and (b) we can generalize to different preference $w$ not seen in training by taking advantage of the shared structure in the noisy manifold. Training the preference-conditioned discriminator model is simple: we can sample from a preference distribution $p(w)$ during training, use $w$ to encode multiple objectives to a single-objective function, and learn a function approximator $f_\theta$.[1]

## 5 Experiment on discrete image generation ($|\mathcal{X}| \approx 10^{236}$)

To validate our method, we compare it against dWJS for the binarized static MNIST image generation task (Salakhutdinov & Murray, 2008; Larochelle & Bengio, 2008). The high-dimensionality of this task ($28 \times 28 \times 2$) makes it an attractive one to validate our approach. For this task, two questions interest us.

- *Does gradient guidance enable more realistic generation than denoising walk?*

- *Can gg-dWJS generate conditionally, i.e., can we generate binarized MNIST images with a specific label (say, 0)?*

In figure 2, we show the results of our experiment. On the left, we compare the samples generated by dWJS, gg-dWJS without denoising walk, and gg-dWJS. Here, for gg-dWJS without denoising walk, we only use $\nabla f_\theta(y_k, c)$ as the gradient for the Langevin MCMC. The samples show that gg-dWJS produces the most realistic and diverse samples, which is further affirmed by the lowest Fréchet inception distance (FID) (Heusel et al., 2017) in table 1. Besides, we experiment with performing the walk step using only gradient-guidance, but we find that it cannot generate high-quality samples. On the right, we show the samples of labels 0, 3, and 8 produced by gg-dWJS. They showcase the method's ability to conditionally generate samples. We report the details related to this experiment in Table 9.

---

[1]Training and evaluation code, training data, and model checkpoints are available at https://github.com/zarifikram/gg-dWJS/.

Table 1: Experiment results for binarized static MNIST image generation. Here, we calculate FID on the test data. Note that by binarizing MNIST images, we lose important pixel information, contributing to the high FID.

| Method | FID $\downarrow$ |
| --- | --- |
| dWJS | 54.62 |
| dWJS w/o denoising walk w/ gradient guidance | 89.42 |
| gg-dWJS | **51.88** |

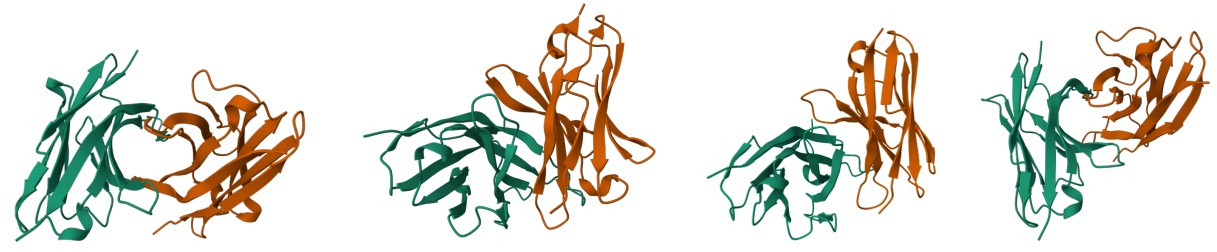

Figure 2: Comparison of binarized MNIST samples generated by different dWJS methods. *Left:* from top to bottom: dWJS, gg-dWJS w/o denoising gradient, gg-dWJS. *Right:* from top to bottom: gg-dWJS generated samples with label 0, 3, and 8.

# 6 Experiment on antibody sequence optimization

## 6.1 Paired chain antibody sequence generation ($|\mathcal{X}| \approx 10^{393}$)

Figure 3: Visualization of generated heavy and light chain sequences using gg-dWJS optimized for instability index. We use IgFold (Ruffolo et al., 2023) for sequence folding and Mol* viewer (Sehnal et al., 2021) for visualization.

Now, we turn our attention to antibody sequence generation. Following Frey et al. (2024); Gruver et al. (2023), we represent antibody sequences as $x = (x_1, \ldots, x_d)$, where $x_i \in 1, \ldots, 21$ refers to the 20 amino acid (AA) type, augmented by the gap token '-'. We use the ∼1.3M sequences provided by Frey et al. (2024). These sequences are from observed antibody space (OAS) database, aligned according to the AHo number scheme (Honegger & PluÈckthun, 2001) using the ANARCI package (Dunbar & Deane, 2016). The gapped heavy and light chain lengths are respectively 149 and 148, making the total dimension $(149 + 148) \times 21$. For the optimization task, we experiment on two simple single-objective tasks: the percentage of beta sheets and the protein instability index (Guruprasad et al., 1990). For each task, we train a smoothed predictor on the antibody sequences and use its gradient guidance to optimize the single objective.

**Results.** We compare gg-dWJS with dWJS, a latent diffusion method generalized for discrete sequences (Kingma et al., 2021, variational diffusion models (VDM)), a transformer-based language model trained for antibody sequence design (Shuai et al., 2023, IgLM), a pre-trained large language model (LLM) (GPT-4o), and a probabilistic method for discrete sequence generation (Bengio et al., 2023; Jain et al., 2022, generative flow networks (GFlowNets)).

Table 2: Statistics of the attribute of the paired OAS dataset.

|  | min | mean | median | max |
|---|---|---|---|---|
| Instability index | 12.687 | 40.400 | 40.618 | 68.230 |
| % Beta sheets | 0.314 | 0.378 | 0.377 | 0.455 |

Table 3: Experiment results for antibody sequence generation for single-objective optimization task. The results show that gg-dWJS-generated sequences are better optimized and of higher quality.

| Method | DCS ↑ | Instability index ↓ | % Beta sheets ↑ |
|---|---|---|---|
| dWJS | $0.49 \pm 0.30$ | $34.14 \pm 6.38$ | $0.393 \pm 0.02$ |
| gg-dWJS w/ Beta sheet discriminator | $0.51 \pm 0.28$ | $35.92 \pm 6.25$ | $\mathbf{0.408 \pm 0.02}$ |
| gg-dWJS w/ Instability discriminator | $\mathbf{0.56 \pm 0.27}$ | $\mathbf{31.32 \pm 5.21}$ | $0.402 \pm 0.023$ |
| VDM | $0.34 \pm 0.31$ | $39.50 \pm 6.79$ | $0.37 \pm 0.02$ |
| IgLM | $0.19 \pm 0.29$ | $38.84 \pm 6.18$ | $0.37 \pm 0.02$ |
| GPT-4o | $0.31 \pm 0.30$ | $42.63 \pm 2.47$ | $0.37 \pm 0.01$ |
| GFlowNets | $0.0 \pm 0.0$ | $39.04 \pm 1.04$ | N/A |

We report the results of our experiments in table 3. Specifically, we report the distributional conformity score (DCS), percentage of beta sheets, and instability index for samples generated by dWJS and gg-dWJS with the two discriminators.[2] We choose the metrics for their fast evaluation following previous works (Stanton et al., 2022; Gruver et al., 2024), though the metrics may have limited impact for real-world designs. We describe the metrics in more details in the Appendix C.3. The results show that with beta sheet gradient and instability index guidance, gg-dWJS generates samples with improved beta sheet percentage (Paired $t$-test, $p$-value $\ll 0.005$) and instability index (Paired $t$-test, $p$-value $\ll 0.005$) than dWJS, respectively. The results also show that our method achieves the best DCS score among all the baselines, showing its capability of generating sequences of high Ablikeness. It is worth noting that we also train the VDM from the same data, but its DCS score is not as high. Besides, our method with a instability index and beta sheet percentage discriminator achieves the best instability index and beta sheet percentage of all methods, respectively. Surprisingly, for the beta sheet percentage counterpart, many other methods come close, but they cannot generate sequences with good ablikeness, as evident by their low DCS. The result shows that not only does gg-dWJS generate antibody sequences that are better optimized for their respective objectives, but it also produces sequences of higher quality overall. Details of this experiment are in appendix C

## 6.2 Antibody CDR H3 design ($|\mathcal{X}| \approx 10^{13}$)

Table 4: Experiment results for antibody sequence generation for single-task optimization task. The results show that gg-dWJS-generated sequences are better optimized and of higher quality.

| Method | $p_{bind}$ | Unique | $E_{dist}$ |
|---|---|---|---|
| gg-dWJS w/ binders only | $0.93 \pm 0.15$ | 0.98 | $5.47 \pm 1.35$ |
| dWJS w/ binders only | $0.87 \pm 0.20$ | 0.99 | $5.98 \pm 1.37$ |
| gg-dWJS | $0.78 \pm 0.33$ | 1 | $6.30 \pm 1.42$ |
| dWJS | $0.53 \pm 0.39$ | 1 | $7.19 \pm 1.34$ |
| VDM | $0.33 \pm 0.35$ | 1 | $7.58 \pm 1.19$ |
| GPT-4o | $0.57 \pm 0.36$ | 1 | $7.42 \pm 1.57$ |
| GFlowNets | $0.32 \pm 0.28$ | 1 | $9.16 \pm 0.78$ |

---

[2]Because of the unavailability of the open-source implementation of DCS by Frey et al. (2024), we implement the metric following the provided algorithm, which produces the same result for dWJS (0.49) reported by the authors.

Now we consider the task of generating CDR mutants on a hu4D5 antibody mutant dataset (Mason et al., 2021). After de-duplication and removal of multi-label samples, this dataset contains 9k binding and 25k non-binding hu4D5 CDR mutants up to 10 mutations. To classify the generated samples, we train a binary classifier that achieves 85% accuracy on a IID validation set. For this task, we generate 1000 samples from each baseline, classify the samples using our trained classifier, and report the binding probability.

**Results.** In table 4, we report the results of the comparison of our method with baselines such as dWJS, GFlowNets, VDM, and GPT-4o. Here, mean binding probability $p_{bind}$ indicates different methods' ability to generate binders. Mean uniqueness and pairwise edit distance $E_{dist}$ indicate the diversity of the sequences. Following Frey et al. (2024), we train our method only on the 9k binders and see a small $p_{bind}$ improvement (0.93) over the base method (0.87). However, this is not ideal because we are not using the full sequence data. Therefore, we train all the baselines on the entire dataset. The results indicate that our method can utilize the sequence data to achieve good diversity (6.3) while having a modest $p_{bind}$ (0.78). Interestingly, GPT-4o achieves a comparable result to dWJS, while comfortably surpassing VDM and GFlowNets. Besides, GFlowNets (top 100) achieves the best diversity with a poor $p_{bind}$. Compared to the methods mentioned above, our method maintains a good balance of diversity and binding probability.

## 6.3 Antibody sequence multi-objective optimization

The main goal of this experiment is to showcase the capability of gg-dWJS to generate samples with preference conditioning. Towards that vision, we choose instability index and beta sheet percentage as the attributes to perform preference conditioning and train the preference-conditioned discriminator $f_\theta(y, w)$ such that $w = \{w_1, 1 - w_1\}$ where $p(w_1) \sim \mathcal{U}(1)$. Followed by a scalar normalization of the attributes, we perform a weighted-sum scalarization to decompose the attributes into a single attribute. Subsequently, using the three different preference vectors $w \in \{[0, 1], [1, 0], [0.5, 0.5]\}$, we generate 1k samples. Figure 5 shows the distribution of the attributes of the generated samples, showing that samples with different preference-conditioning indeed fall into a distinct distribution. Besides, the generated samples exhibit good diversity, as shown in table 5. Finally, in figure 4, we show a pareto front for the generated samples, showcasing our method's ability to perform in a multi-objective setting. The results show that our method is able to generate samples according to different preferences $w$, e.g., $w = [0, 1]$ generates samples focusing on the beta sheet percentage while $w = [1, 0]$ generates samples focusing on the instability index.

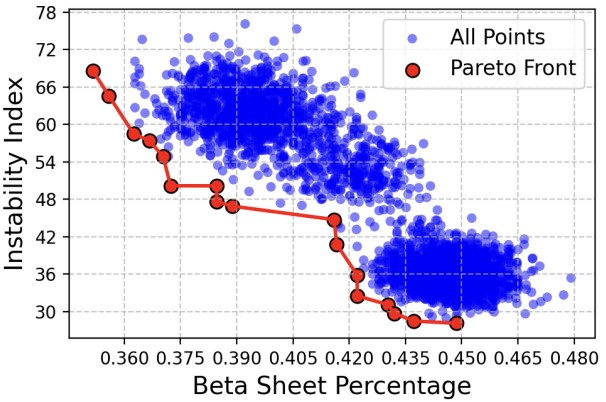

Figure 4: Pareto front of the samples generated using three preference weights with gg-dWJS.

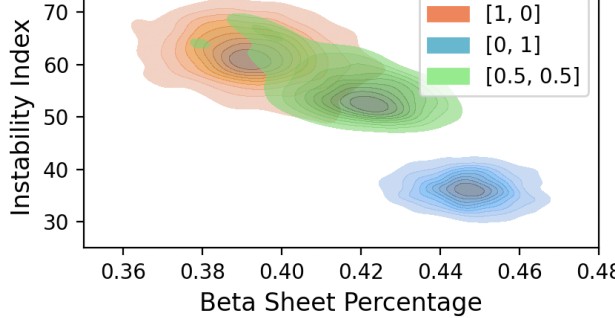

Figure 5: Distribution of the generated samples with our method for three preferences.

|  | $E_{dist}$ ↑ | DCS ↑ |
|---|---|---|
| $[1, 0]$ | $41.03 \pm 13.26$ | $0.29 \pm 0.13$ |
| $[0, 1]$ | $21.27 \pm 5.5$ | $0.05 \pm 0.13$ |
| $[0.5, 0.5]$ | $51.16 \pm 31.79$ | $0.10 \pm 0.08$ |

Table 5: Experiment results for preference conditioning with gg-dWJS for antibody sequence optimization for three different preferences. The results show that samples generated with preference conditioning using gg-dWJS are diverse.

## 6.4 Effect of guidance strength

To understand the effect of the relative gradient-guidance strength $\lambda$, we run the antibody sequence optimization with the instability index discriminator guidance for $\lambda \in \{10^0, 10^1, 10^2, 10^3\}$ while keeping $K$ fixed at 40. Figure 6 shows the result of our experiment, showing that optimization and diversity are in tension—greater optimization leads to sampling from a smaller region of the sampling space. Similarly, we can see that higher guidance strength only leads to improvement in DCS up to a certain extent, after which the DCS reduces.

Table 6: Ablation results on $\lambda$ for antibody sequence optimization task. The results show that while higher gradient-guidance strength leads to better optimization, it also leads to less diversity and data fidelity.

| $\lambda$ | Instability index $\downarrow$ | DCS $\downarrow$ | $E_{dist} \uparrow$ |
|---|---|---|---|
| $10^0$ | $31.32 \pm 5.20$ | $0.56 \pm 0.27$ | $89.07 \pm 26.80$ |
| $10^1$ | $26.29 \pm 4.04$ | $0.57 \pm 0.27$ | $75.90 \pm 29.89$ |
| $10^2$ | $22.16 \pm 3.63$ | $0.58 \pm 0.27$ | $34.04 \pm 24.11$ |
| $10^3$ | $16.43 \pm 3.42$ | $0.51 \pm 0.28$ | $28.79 \pm 9.79$ |

## 7 Experiment on anti-microbial peptide (AMP) design

The goal of this experiment is to generate peptides with anti-microbial properties. To this end, we collect AMPs and non-AMPs from the DBAASP database (Pirtskhalava et al., 2021). Following (Jain et al., 2022), we choose peptides of sequence length of 12 to 60 with the target group Gram-positive bacteria, which yields 6438 positive AMPs and 9222 non-AMPs.[3] Then, we split the dataset into two parts: $D_1$ and $D_2$, following Angermueller et al. (2019) where $D_1$ is visible to our algorithm and $D_2$ is used to train the oracle for validation of the results. To split the dataset, we follow the same principle as Jain et al. (2022): for any peptide $x \in D_1$, there are no peptides $x' \in D_2$ such that $x'$ belongs to $x$'s group and vice versa. This split yields 3219 AMPs and 4611 non-AMPs in $D_1$. Finally, we train the oracle utilizing ProtTrans (Elnaggar et al., 2007) features from $D_2$ and utilizing MLP classifiers (89% accuracy). For the gg-dWJS implementation, we follow the same score model and discriminator model setup from the antibody CDR H3 design experiment (details in Sections D.1 and D.2). Finally, to adapt to the variability of sequence length, we pad all sequences with *gap tokens* appended to the end.

**Results.** We compare our method against baselines such as GFlowNet-AL (Jain et al., 2022), DynaPPO (Angermueller et al., 2019), COMs (Trabucco et al., 2021), and GFlowNets (Bengio et al., 2021) utilizing the reported data from Jain et al. (2022) under the criteria *performance, diversity, and novelty*. Here, performance measures the count of samples classified as AMP by the oracle, diversity measures the difference among the generated samples, and novelty measures the difference between the generated samples and the training dataset ($D_1$). We detail the metrics in Appendix C.3.

Table 7: Results on the AMP Task.

| | **Performance** | **Diversity** | **Novelty** |
|---|---|---|---|
| **gg-dWJS** | $\mathbf{0.98 \pm 0.015}$ | $\mathbf{25.78 \pm 1.22}$ | $15.021 \pm 1.02$ |
| **GFlowNet-AL** | $0.932 \pm 0.002$ | $22.34 \pm 1.24$ | $\mathbf{28.44 \pm 1.32}$ |
| **DynaPPO** | $0.938 \pm 0.009$ | $12.12 \pm 1.71$ | $9.31 \pm 0.69$ |
| **COMs** | $0.761 \pm 0.009$ | $19.38 \pm 0.14$ | $26.47 \pm 1.3$ |
| **GFlowNet** | $0.868 \pm 0.015$ | $11.32 \pm 0.67$ | $15.72 \pm 0.44$ |

Table 7 reports the results. We see that gg-dWJS achieves significantly better performance and diversity than the baselines. Besides, our method reports the results for 100 generated samples over 10 random seeds, while the baselines only report results for the TopK samples from $t * K$ generated samples, where

---

[3]In Jain et al. (2022), it is mistakenly reported as 9522 non-AMPs.

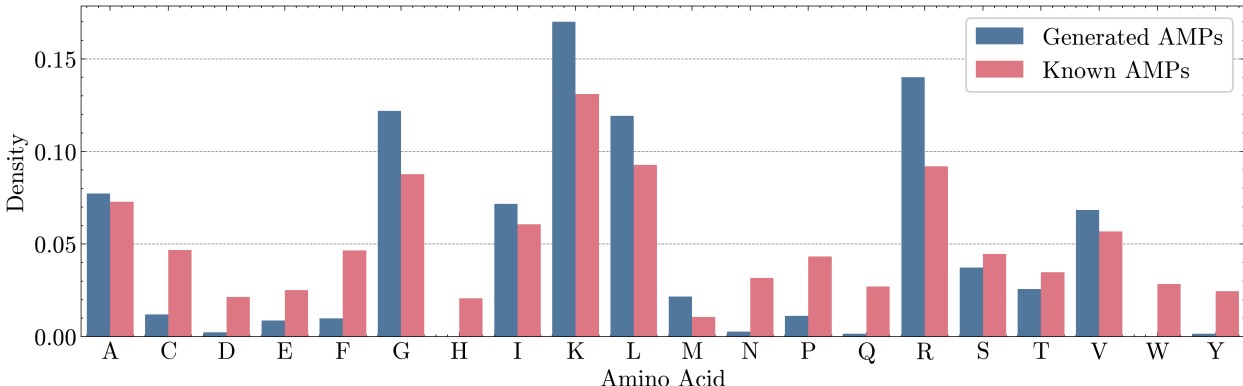

Figure 6: Distribution of amino acids found in the generated AMPs by gg-dWJS matches that of known AMPs while maintaining focusing on amino acid "K", which is dominant in peptides with anti-microbial activity.

$K = 100$ and $t = 5$. However, it is also evident that GFlowNet-AL achieves better novelty than our method. However, as Kirjner et al. (2023) reports, higher novelty is *not necessarily equivalent* to better results. For instance, a random algorithm would achieve maximum novelty.

We also see a lower average instability index of 26.3 compared to 26.5 reported by GFlowNet-AL, showing that gg-dWJS generates biologically relevant sequences (score of over 40 indicates instability). Figure 6 shows comparison of the distribution of amino acids between the known AMPs and gg-dWJS generated AMPs. We can see that gg-dWJS generates peptides that retains the amino acid profiles of the known AMPs but boosts important characteristics generally found in AMPs, e.g., the amino acid "K", which is dominant in peptides with anti-microbial activity (Jain et al., 2022).

## 8 Related Work

### 8.1 Discrete generative models

A large class of discrete generative models consist of autoregressive models (i.e., language models). Among many others, Austin et al. (2021) learns the posterior alphabet distribution over the prior generated data by learning the bidirectional context in a language model. Recent works such as Generative Flow Networks (GFlowNets) (Bengio et al., 2023; 2021) model the flow as a DAG, and learn to sample directly proportional to a given reward function. Zhang et al. (2022) uses this idea to train and sample from an energy based model (EBM) (LeCun et al., 2006) using contrastive divergence (Carreira-Perpinan & Hinton, 2005). Besides, Grathwohl et al. (2021) improves the sampling process by leveraging local gradients using a Gibbs sampler (George & McCulloch, 1993; Gelfand, 2000).

Another way to generate data is through denoising models (Ho et al., 2020; Song & Ermon, 2020; Song et al., 2021). These models learn the gradient log density of the data and generate continuous data from the perturbed data distribution. Thus, they are faster and more efficient than autoregressive models. Of course, one cannot simply apply the denoising gradient to the discrete data distribution because the gradients are not defined there. Many promising works attempt to remedy this by proposing different techniques. Chen et al. (2023) simply transforms the discrete data into analogue bits, learns the denoising gradient, and applies thresholding to return the categorical data. Sun et al. (2023) applies denoising via a continuous-time Markov chain to the categorical data using a stochastic jump process. For graph adjacency matrices, Niu et al. (2020) adds Gaussian perturbation to the upper triangular matrix, sample using the learn score model, and transform samples in the continuous space by quantizing the generated continuous adjacency matrix to a binary one. On a similar vain, Yan et al. (2024) performs post-hoc thresholding after the end of Langevin dynamics to tranform the denoised continuous adjacency matrix to a binary or discrete adjacency matrix.

Frey et al. (2024) leverages the gradient information in the smoothed data manifold and jumps to the true data manifold using NEB, which allows obtaining a sample at any part of the Langevin dynamics.

## 8.2 Protein sequence optimization

With the advancement of language models, many recent works (Madani et al., 2020; Nijkamp et al., 2023; Ferruz et al., 2022) propose language models pre-trained on protein data to generate protein sequences, often for tasks such as structure prediction (Lin et al., 2023). These protein language models have also been adopted for antibody generation tasks, both unguided (Shuai et al., 2021) and guided (Gligorijević et al., 2021; Ferruz & Höcker, 2022; Tagasovska et al., 2022). A key problem with language modeling of antibodies is that they struggle to capture the data distribution of the antibody sequences given a limited amount of high-quality data and high-entropy variable regions of the antibody sequences. As such, recent works such as Luo et al. (2022); Gruver et al. (2023); Peng et al. (2023) leverage the continuous space of antibody structures to generate antibody structures using diffusion. In this work, we focus on the problem of generating targeted antibody sequences.

## 9 Conclusion and limitations

In this work, we introduce gg-dWJS, which learns a discriminative model on the smoothed data and uses the gradient information to augment its denoising walk towards the local maxima given some attributes. Finally, it returns to the clean data manifold by using a conditional jump with the same model. Thus, our method requires no additional score model training for different optimization tasks. Additionally, we formalize our method in a multi-objective setting using a preference-conditional discriminator model. Using preference-conditioning, we show that our method can generate samples that correspond to different regions of the multi-objective space based on preference. We show our method's applicability in two modalities: discrete imagine generation and biological sequence generation in single objective and multi objective cases.

Despite our sincere efforts, our work falls short in addressing numerous issues and has limitations. For starters, while we see an improvement in DCS with gradient guidance, the improvement starts to fade away with the higher strength of the gradients. It is also worth mentioning that more optimization leads to less diversity in the samples. Indeed, the general tension of data fidelity, diversity, and optimization still holds true in our case, and improving them together is still a challenge. For example, our current approach with MCMC can be a limitation for diversity, as it is known to suffer from mode collapse, where it diverges to a local solution for many starting points. There are many line of work that attempts to tackle this problem (De Souza et al., 2022; Liu et al., 2023; Wang et al., 2024b) from different viewpoints, which we believe can be an useful addition to our work. Future works can also tackle the problem of optimization and uniqueness together through the addition of an active learning pipeline (Settles, 2009; Hernandez-Garcia et al., 2023) to our method. Besides, one can improve fidelity of the generated sequences through the use of strcture priors, similar to the recent work by Wang et al. (2024a). Furthermore, we did not investigate different scalarization functions in our preference-conditioning setting; we left that along with the multi-objective benchmarking of our method for future work.

**Broader Impact Statement**

This work falls in the line of discrete generative modeling. While the negative impacts of generative modeling apply to it, the authors do not foresee any particular negative impacts.

**Acknowledgments**

This work is funded by RISE student research grant (S2024-01-013) and NUS-UToronto joint grant. Furthermore, Z. Ikram is thankful to his family.

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

*Supplementary material for*

# Gradient-guided discrete walk-jump sampling for biological sequence generation

Zarif Ikram, Dianbo Liu, M Saifur Rahman

## Acronyms

**AA** amino acid. 7

**DAG** directed acyclic graph. 11

**DCS** distributional conformity score. 8–10, 19, 20, 23

**dWJS** discrete walk-jump sampling. 1–4, 6–8, 21

**EBM** energy based model. 11

**FID** Fréchet inception distance. 6, 19

**GFlowNets** Generative Flow Networks. 11

**gg-dWJS** gradient-guided discrete walk-jump sampling. 1, 2, 4–8, 12, 21, 23, 24

**GRAVY** grand average of hydropathicity. 19

**KDE** kernel density estimation. 19

**MCMC** Markov chain Monte Carlo. 1–4

**MOO** multi objective optimization. 20, 21

**NEB** Neural empirical Bayes. 2, 3, 5, 12

**OAS** observed antibody space. 7, 19–21, 23

## A    Notations

Table 8: Notations summary

| Symbol | Description |
| --- | --- |
| $X$ | discrete random variable |
| $Y$ | smoothed random variable |
| $X \rightharpoonup Y$ | kernel density estimation smoothing |
| $Y \rightharpoonup X$ | denoising mechanism of empirical Bayes |
| $g_\phi(y)$ | learnt score function |
| $f_\theta(y)$ | learnt discriminator function |
| $w$ | preference condition weights |
| $\delta$ | step size |
| $u$ | inverse mass |
| $\gamma$ | friction |
| $\lambda$ | relative gradient-guidance strength |
| $K$ | number of steps taken |

# B    Additional details on the static MNIST experiment

We train the noisy score and discriminator model on the binarized static MNIST dataset.[4]  For score model architecture, we use a U-Net architecture (Ronneberger et al., 2015) that takes the smoothed one-hot representation of the discrete images and returns the score of the same shape.  Finally, we train a CNN architecture [5] on smoothed data and their corresponding labels for the discriminator model.

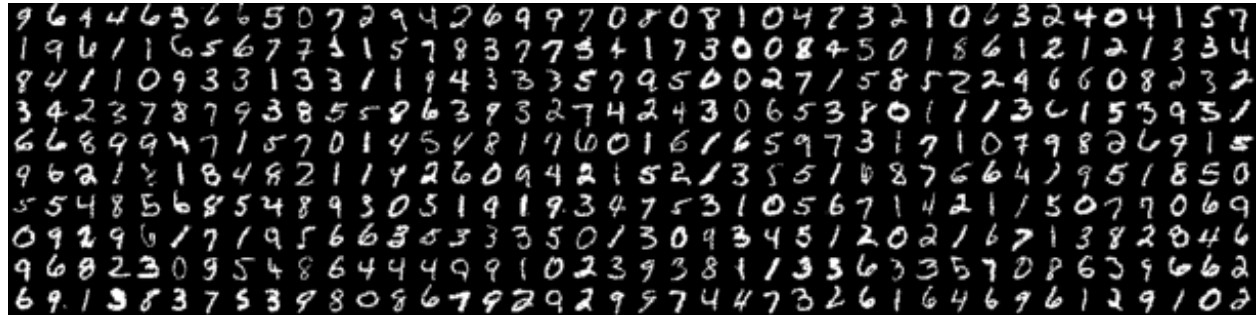

Figure 7: Randomly chosen 400 static MNIST test-set samples.

# C    Additional details on the antibody sequence optimization experiment

## C.1    Training details

All our models are trained with maximum 40 epochs with early stopping and with a learning rate of $10^{-4}$, a weight decay of 0.01, and a batch size of 64. We validate our models on a held-out validation set separate from the training data. Both the discriminator and the score based models for all experiments are trained with mean square error loss, except for CDR H3 experiment, in which the discriminator is trained with a binary cross entropy loss. We save the best models based on the validation loss and use the model during sampling.

## C.2    distributional conformity score (DCS)

The recently proposed DCS measures sample quality using a sample-to-distribution metric, as opposed to previous sample recovery metrics (Jin et al., 2021).  It measures the fraction of the validation sequences that are less *similar* to their IID samples compared to a target sample. We show the algorithm to calculate DCS from Frey et al. (2024) in algorithm 2.  We implement DCS using two sequence-based properties: molecular weight and grand average of hydropathicity (GRAVY) (Kyte & Doolittle, 1982). We used the kernel density estimation (KDE) implementation by Scikit-learn (Pedregosa et al., 2011) to approximate the joint distribution using a Gaussian kernel and a bandwidth of 0.15. DCS is a data-fidelity metric similar to FID for images. Unsurprisingly, DCS and optimized attributes are in tension—optimization while preserving data fidelity is harder.

## C.3    Evaluation metrics

**Paired OAS experiment.**    We show the optimization of two attributes in this experiment: % beta sheets and instability index. The fraction of beta-pleated sheets provides insights into the structural characteristics of antibody sequences, influencing their stability and functionality.  The protein stability index is a test proposed by Guruprasad et al. (1990). A protein sequence has a short half-life if its stability index is above 40. Stability is a crucial metric in the context of therapeutic antibody generation, as it is closely associated with manufacturability.

---

[4]Collected from Kaggle

[5]Based on the PyTorch example, adapted for discrete data.

---

**Algorithm 2** distributional conformity score (DCS) (adapted from Frey et al. (2024))

---

**Input:**

$\mathcal{D}$: reference distribution

$x \in \mathcal{X}$: test example

$A$: conformity measure

**Output:**

$p^y$: the fraction of validation examples that are less similar to $\mathcal{D}_{z|y}$ than $x$

Sample $(z_1, \ldots, z_n)$, $z_i \sim \mathcal{D}_{z|y}$ and a held-out validation set $(\tilde{z}_1, \ldots, \tilde{z}_{k-1})$, $\tilde{z}_i \sim \mathcal{D}_{z|y}$

$\tilde{z}_k \leftarrow (x, y)$

**for** $i = 1, \ldots, k$ **do**

$\quad \lfloor \quad \alpha_i \leftarrow A(z_1, \ldots, z_n, \tilde{z}_i)$

$p^y \leftarrow \frac{1}{k} \sum_{i=1}^{k} [\alpha_i < \alpha_k]$

**return** $p^y$

---

**CDR H3 experiment.** The key attribute for this experiment is $p_{bind}$, the probability of being a binder, which we obtain from the binary classifier trained on the data. Besides, to measure the diversity of the generated sequences, we use mean uniqueness and edit distance $E_{dist}$.

**Paired OAS MOO experiment.** The metrics used in this experiment are similar to those listed above: % beta sheets and instability index for the optimization metric, edit distance for the diversity metric, and DCS for the data fidelity metric.

**AMP experiment.** Following Jain et al. (2022), we use the evaluation metrics described below. Here, $\mathcal{D}$ is the dataset of generated samples, $\mathcal{D}_0$ is the dataset used to train the generative models, and $\mathcal{X}$ is the set of all peptides.

- **Performance**: This is simply the percentage of the generated samples identified as AMPs by the oracle trained on $D_2$, favoring models that can consistently generate more AMPs.

- **Diversity**: In addition to being high performance, it is important that samples generated by the models are diverse. This diversity can be quantified by measuring the distance between all the generated samples, i.e.,

$$\text{Diversity}(\mathcal{D}) = \frac{\sum_{x_i \in \mathcal{D}} \sum_{x_j \in \mathcal{D} \setminus \{x_i\}} d(x_i, x_j)}{|\mathcal{D}|(|\mathcal{D}| - 1)} \tag{6}$$

where $d$ is the Levenshtein distance (Levenshtein, 1966) calculated with the `Polyleven` package.

- **Novelty**: Novelty seeks to measure the difference between the proposed candidates $\mathcal{D}$ and the candidates that are already known $\mathcal{D}_0$. We measure this in the proposed candidates as follows:

$$\text{Novelty}(\mathcal{D}) = \frac{\sum_{x_i \in \mathcal{D}} \min_{s_j \in \mathcal{D}_0} d(x_i, s_j)}{|\mathcal{D}|} \tag{7}$$

where $d$ is the Levenshtein distance.

# D   Implementation details of the baselines

## D.1   Score models

We follow Frey et al. (2024) for the score model implementation available at `https://github.com/Genentech/walk-jump`. We use a 35-layer Bytenet (Kalchbrenner et al., 2016) architecture with a hidden layer of 128. To train the model from scratch, we utilize a batch size of 64 and

the AdamW optimizer (Loshchilov & Hutter, 2019) in PyTorch (Paszke et al., 2019) with early stopping. The training parameters include a learning rate of $10^{-4}$ and a weight decay of 0.01. We conducted the training using four NVidia A100 GPUs. The training process takes approximately 10 hours. For training the score model, we use $\sigma = 1$. We use the same score model training method for all experiments in this paper. Table 9 lists all the hyperparameters we use for sampling from the score model using dWJS and gg-dWJS.

Table 9: Hyperparameters used for dWJS and gg-dWJS sampling.

| | |
|---|---|
| $\sigma$ | 1 |
| $\delta$ | 0.5 |
| $\gamma$ | 1 |
| $K$ | 40 |
| $\lambda$ (% beta sheet) | 100 |
| $\lambda$ (instability index) | 1 |

### D.2    Discriminator models

**Paired OAS experiment.**   We train the discriminator model on two sequence-related properties: the percentage of beta sheets and the protein instability index. We label the training sequences using BioPython (Cock et al., 2009). For the architecture of the smoothed predictor, we adopt a 3-layer 1D-CNN followed by a 3-layer MLP integrated into the existing ByteNet architecture, incorporating leakyReLU (Xu et al., 2020) activations between the layers. Finally, we follow the same training parameters as the score model training. To use the discriminator models' prediction gradients as guidance, we use the `torch.autograd` on the models prediction given a label or not (in case of a non-categorical variable).

**CDR H3 experiment.**   Here, we have two similarly trained discriminator models for two different purposes: the discriminator model on $p_b ind$ for classification and the noised discriminator model for generation using gg-dWJS. We adopt a similar approach to the one used in the paired OAS experiment, except for using a sigmoid terminal activation and binary cross entropy loss function to accommodate the binary classification.

**Paired OAS MOO experiment.**   The chosen attributes for the MOO experiment are instability index and % beta sheets. For this experiment, we attempt to lower both values as the multi-objective criteria. Therefore, we train a noised discriminator model that outputs a weighted sum of the normalized attributes. To achieve preference conditioning, the setup is similar to the other discriminators, except we concatenate the preference vectors with the output of the 1D-CNN.

### D.3    Variational diffusion models (VDM)

In this baseline, we project the discrete sequences into a continuous latent space and perform the diffusion in the latent space. We use the implementation from `https://github.com/addtt/variational-diffusion-models` to train the VDM model on the OAS dataset by performing gradient descent on the reconstruction and diffusion loss. We use a generalization of the same model to sample discrete sequences by performing a reverse diffusion process for 100 steps.[6] Our implementation uses hyperparamers as listed in the table 10.

### D.4    IgLM

IgLM is the autoregressive baseline for our comparison. We use the open-source package from `https://github.com/Graylab/IgLM` to sample heavy and light chain samples. We used the following prompt to generate the heavy sequences, with an adjustment to the `chain_token` attribute to generate the light sequences.

---

[6]In layman's terms, we perform an `argmax` on the last step while sampling.

Table 10: Hyperparameters used in the VDM implementation.

| | |
|---|---|
| Embedding dimension | 128 |
| Attention head | 1 |
| Dropout probability | 0.1 |
| $\gamma_{min}$ | -13.3 |
| $\gamma_{max}$ | 5 |
| Learning rate | $2 \times 10^{-4}$ |
| Weight decay | 0.01 |

```
METHOD_NAME = "IgLM"
prompt_sequence = ""
chain_token = "[HEAVY]"
species_token = "[HUMAN]"
num_seqs = 1000
```

## D.5 GPT-4o

Following Frey et al. (2024), We use GPT-4o as a large language model (LLM) baseline. For the paired OAS generation experiment, we prompt the following prompt.

```
You are an expert antibody engineer.  I am going to give you examples of antibody
heavy chain and light chain variable regions from the paired observed antibody
space database.  You will generate 10 new antibody heavy chain and light chain that
are not in the database.  Output the 10 samples as a python list.  Here are the
examples:
[('QVQLVQS-GTEVKKPGSSVKVSCKASG-GTFSS---Y ... DYYCQAWDY-----------STAVFGTGTKVTVL')]
```

Similarly, we perform a similar prompt for the CDR H3 design experiment. The prompt is the following:.

```
You are an expert antibody engineer.  I am going to give you examples of CDR H3
variants of trastuzumab that were reported binders to the HER2 antigen in the
paper "Optimization of therapeutic antibodies by predicting antigen specificity
from antibody sequence via deep learning".  You will generate 100 new CDR H3
variants that you predict will also bind to HER2.  Output the 100 samples as a
python list.  Do not post the code.  Instead, show the samples directly.  Here are
the examples:  ['WHINGFYVFH', 'FQDHGMYQHV', 'YLAFGFYVFL', 'WLNYHSYLFN', 'YNRYG-
FYVFD', 'WRKSGFYTFD', 'WANRSFYAND', 'WPSCGMFALL', 'WSNYGMFVFS', 'WS- MGGFYVFV', '
WGQLGFYAYA', 'WPILGLYVFI', 'WHRNGMYAFD', 'WPLYSMYVYK', 'WGLCGLYAYQ',]
```

## D.6 Generative flow networks (GFlowNets)

GFlowNets is a popular baseline for discrete generative modeling using a probabilistic method. However, for the current implementation of GFlowNets, generating samples in a state space of size $21^{249}$ is difficult. Therefore, we reduce the problem of generating the paired sequences to generating the first 50 mutations of the full sequences with trajectory balance objective (Malkin et al., 2022). The heavy and light chain suffixes added to the generated mutations are QVQLVQSGTEVKKPGSSVKVSCKASGGTFSSYAVSWVRQAPGQ-GLEWMGR FIPILNIKNYAQDFQGRVTITADKSTTTAYMELINLGPEDTAVYYCARGSLSGREGLP-LEYWGQGTLVSVSS and EVVMTQSPATLSVSPGESATLYCRASQIVTSDLAWYQQIPGQAPRLLI-FAASTRATGIPARFSGSGSETDFTLTISSLQSEDFAIYYCQQYFHWPPTFGQGTKVEIK, respectively. We use the implementation provided in Bengio et al. (2023) for our experiment. We use a 3-layer MLP with

256 hidden dimensions to model the forward layer, which is then followed by a leaky ReLU. The forward layer takes the one-hot encoding of the states as inputs and outputs action logits. To model the forward and backward flow, we double the action space and train the MLP. with a learning rate of $10^{-3}$, including a learning rate of 0.1 for $Z_\theta$. Since GFlowNets can learn to sample proportionally to a reward distribution, we transform the optimization criteria as a reward function $\mathcal{R}(x)$ in the following way:.

**Paired OAS experiment: optimizing instability index.** We label the training sequences using BioPython (Cock et al., 2009) and transform the instability index to a reward by performing the following transformation: $R(x) = 2^{\frac{index-5}{10}}$ where $index$ is the instability index of $x$.

**CDR H3 experiment: optimizing binding likelihood.** We use the training classifier used for evaluation to get the binding likelihood of the training sequences. The reward transformation in this task is $R(x) = 2^{p_{bind}} - 1$.

# E  Additional experiments

## E.1  Effect of $K$

To understand the effect of the number of sampling steps $K$, we perform an ablation experiment on $K$ using the instability index discriminator model as the gradient guidance. For this experiment, we change $K \in \{10, 20, 30, 40, 50\}$ while keeping $\lambda = 1$. Table 11 shows the results of the experiment. We can see that there is a minimal change in the attribute, data fidelity, and diversity. In a close inspection, we see that as $K$ increases, sampled sequences are more optimized, yet less diverse. Notably, DCS continues to improve until $K = 40$, after which it decreases.

Table 11: Ablation results on number of samples steps $K$ for antibody sequence optimization task. We see a generally indifferent yet a slightly optimized attribute in exchange of a reduced diversity as $K$ increases.

| $K$ | Instability index | DCS | $E_{dist}$ |
|---|---|---|---|
| 10 | $31.69 \pm 5.42$ | $0.52 \pm 0.28$ | $95.35 \pm 24.35$ |
| 20 | $31.96 \pm 5.25$ | $0.53 \pm 0.28$ | $91.23 \pm 25.47$ |
| 30 | $31.41 \pm 5.42$ | $0.54 \pm 0.27$ | $90.34 \pm 26.55$ |
| 40 | $31.32 \pm 5.20$ | $0.56 \pm 0.27$ | $89.07 \pm 26.80$ |
| 50 | $31.38 \pm 5.33$ | $0.55 \pm 0.27$ | $87.47 \pm 26.70$ |

## E.2  Statistical analysis on preference conditional generation

To determine the statistical significance of the samples generated with different preferences using gg-dWJS from figure 5, we run a Friedman test (Friedman, 1937; 1940).[7] In this case, we utilize three distinct preferences to represent three different treatments. Table 12 shows the result of the test. The table contains three main attributes: the proportion of beta sheets and the instability index, each analyzed using different weight combinations. The Friedman test yields low p-values ($p < 0.001$), indicating statistically significant differences between weight arrangements. For example, sequences optimized with w1=1 and w2=0 have a beta sheet percentage of $0.44 \pm 0.01$, whereas those with w1=0 and w2=1 have a greater instability index of $61.84 \pm 4.23$. These results demonstrate our method's capacity to build sequences based on certain optimization preferences.

## E.3  Experiments with more objectives

To understand the capability of our method under more objectives, we conduct experiments with 5 objectives: target molecular weight, inverse sequence length, negative GRAVY, % beta sheets, and inverse instability index, with our preference weight $w$ indictating their preference in that order. We set the target molecular

---

[7]We use the SciPy package (https://scipy.org).

Table 12: Experiment results for antibody sequence generation for multi-objective optimization task. Here the two objectives are percentage of beta sheets and aromaticity. The results show that gg-dWJS-generated sequences are optimized for different preferences. We use Friedman test to demonstrate that the attributes are statistically different.

|  | statistic | p-value | w1=1,w2=0 | w1=0,w2=1 | w1=0.5,w2=0.5 |
|---|---|---|---|---|---|
| % Beta sheet | 1564 | <0.001 | $0.44 \pm 0.01$ | $0.39 \pm 0.01$ | $0.43 \pm 0.02$ |
| Instability index | 1522 | <0.001 | $36.11 \pm 2.45$ | $61.84 \pm 4.23$ | $41.90 \pm 8.54$ |

weight to 23000 and use $e^{-(\texttt{molWt}-23000)^2/23000}$ as our objective which is maximized at 23000. Table 13 shows the results of our experiment. We can see that with the objectives are satisfied with their corresponding preferences. For example, with $w = [0, 0, 0, 0, 1]$ the instability index is minimized while with $w = [1, 0, 0, 0, 0]$, the molecular weight is closer to the target weight.

Table 13: Experiment results for antibody sequence generation for multi-objective optimization task for five objectives.

| w | Molecular weight | Sequence length | GRAVY | % Beta sheets | Instability index |
|---|---|---|---|---|---|
| $[1,0,0,0,0]$ | $\mathbf{23915.45 \pm 475.02}$ | $231.36 \pm 3.68$ | $-0.25 \pm 0.08$ | $0.39 \pm 0.02$ | $36.40 \pm 6.14$ |
| $[0,1,0,0,0]$ | $24727.72 \pm 454.44$ | $\mathbf{229.23 \pm 3.53}$ | $-0.26 \pm 0.09$ | $0.39 \pm 0.02$ | $35.63 \pm 6.26$ |
| $[0,0,1,0,0]$ | $24796.43 \pm 472.75$ | $229.49 \pm 3.73$ | $\mathbf{-0.28 \pm 0.09}$ | $0.39 \pm 0.02$ | $35.25 \pm 6.06$ |
| $[0,0,0,1,0]$ | $24942.65 \pm 483.04$ | $230.78 \pm 3.52$ | $-0.24 \pm 0.08$ | $\mathbf{0.41 \pm 0.02}$ | $32.04 \pm 4.71$ |
| $[0,0,0,0,1]$ | $25273.25 \pm 373.58$ | $231.70 \pm 3.24$ | $-0.25 \pm 0.10$ | $0.39 \pm 0.02$ | $\mathbf{26.95 \pm 3.89}$ |

## F  Discussion

*Why does discriminator gradient guidance produce higher quality samples?* We speculate that the guidance assists the denoising walk by reducing the discrete space. Moreover, by collaborating with the denoising score, it strengthens the actions taken in relation to a particular label, thus enhancing the accuracy of the generated data. There are many works, such as Dhariwal & Nichol (2021); Kawar et al. (2022), that report the same phenomenon. We leave the experimental validation of this underlying effect for future work.

