# OpenReview forum: "Gradient-guided discrete walk-jump sampling for biological sequence generation"
_TMLR — Accepted by TMLR_

### Review · Reviewer_bSDE · 2024-10-07

**Summary Of Contributions:**

This work presents gradient-guided discrete walk-jump sampling (gg-dWJS), an extension of guiding dWJS towards different properties, and applies it to antibody sequence optimization. The extension is fairly straightforward by adding the score of a classifier during dWJS sampling. To perform multi-objective optimization, multiple scores are combined with Pareto optimal scalarization. The method is first demonstrated on binary MNIST before moving to antibodies benchmarks following the original dWJS work. Two guidances are demonstrated: instability index and beta sheet. The results show there are improvements on both fronts though beta sheet improvement is minimal and the significance of the instability index improvements is also small. The baselines compared to include GPT-4o, GFlowNets, and VDM.  Ablations are performed on guidance parameters.

**Audience:**

No

**Broader Impact Concerns:**

No concerns.

**Claims And Evidence:**

No

**Requested Changes:**

* Technical details need to be clarified such as how the discriminators were trained and how their predictions were used to parameterize the gradient guidance scores.
* $\lambda$ is referenced in the experiments as the guidance weight but is not specified in the methods.
* What is inverse mass and friction?
* Please explain the significance of the improvements achieved with gradient guidance. Why should protein engineering or discrete optimization community care about the improvements?
* My recommendation would be to find discrete optimization tasks with established benchmarks such as the original gflownets paper. There needs to be stronger benchmarks and experiments since the technical novelty of this work is limited. Another issue is the baselines such as GPT-4o or VDMs which are not meant for biological problems.




[1] https://proceedings.mlr.press/v162/jain22a/jain22a.pdf

**Strengths And Weaknesses:**

Strengths:
* Gradient guidance is a simple extension of dWJS that is easy to apply.
* Evaluating gg-dWJS to multi-objective optimization is an important experiment that I'm glad was included.

Weaknesses:
* Due to gradient guidance being simple, we expect strong experiments or analysis of the technique. Gradient guidance is not a novel idea since it has been used in classifier guidance [1], Gibbs with gradients [2, 3]. Unfortunately, the antibody experiments don't present a compelling reason to use this technique.
* The beta sheet guidance is especially puzzling since the base distribution should already capture the desired beta sheet composition. The improvement in beta sheet is 0.408 compared to 0.393 which is insignificant. There is little justification of why we care about this guidance.
* Instability is a property I have not seen before for antibody optimization. The significance of the improvement, 31.3 compared to 34.1, is once again unclear.
* Technical and experiment details are missing such as how the classifiers were trained. It is also unclear how a continuous valued property (beta sheet percentage, instability index) was utilized as a discriminator in gradient guidance. I have looked in the appendix and could not find these details. Hyperaparameters such as inverse mass, friction in algorithm 1 are not explained either.


References:
[1] https://arxiv.org/abs/2105.05233
[2] http://proceedings.mlr.press/v139/grathwohl21a/grathwohl21a.pdf
[3] https://arxiv.org/abs/2406.01572v1

---

> ### Author Response · Authors · 2024-10-21
> **Reply to the reviewer bSDE**
>
> Dear reviewer bSDE,
>
> The insights you provided in the reviews are invaluable, and we would like to thank you for your time crafting this review. Please go through the changes we made in blue in the updated manuscript.
>
> In what follows, we address your comments one by one.
>
> ----------
>
> **RC1. Missing details about technical details (e.g., discriminator training, gradient scores)**
>
> Thank you for your suggestion. As suggested, we have revised our work and added more technical details in the **Appendix C1 and D2**.
>
> ----------
>
> **RC2. Missing guidance weight $\lambda$ in the methods.**
>
> Thank you for pointing it out correctly. We have corrected it in **Section 3.3**.
>
> ----------
>
> **RC3. Missing details about inverse mass and friction.**
>
> We have followed the previous works [1, 2] for the algorithm. We have cited the original papers and added the hyper-parameters in the **Section 3.3**.
>
> ------------
> **RC4. Missing motivation for significance of the improvements achieved with gradient guidance**
>
> We added paired t-test results to indicate statistical significance in **Section 6.1**. Besides, we also talk in details about why discrete optimization is important in the **Section 1**. It is without a doubt that the metrics we benchmarked our results with has a little involvement in the protein design community. We chose such metrics because they are easy to calculate. The key claim of our work lies on the fact that our method can be utilized to produce samples that show good fidelity (DCS score/FID score) with targeted optimization (label conditioning in discrete MNIST/optimized antibody sequences for the chosen metrics).
>
> ----------
>
>
>
>
>
> **RC5. More discrete optimization tasks with established benchmarks such as the original gflownets paper**
> In the original GFlowNets paper, the focus is on *de novo* generation, where there is a reward function from which we want to sample from. Here, we focus on *ab initio*, where we have a dataset of samples, and we want to generate similar but unique samples.
>
> ------------
> Refereneces
> 1. Frey, N. C., Berenberg, D., Zadorozhny, K., Kleinhenz, J., Lafrance-Vanasse, J., Hotzel, I., ... & Saremi, S. Protein Discovery with Discrete Walk-Jump Sampling. In The Twelfth International Conference on Learning Representations.
> 2. Saremi, S., & Hyvärinen, A. (2019). Neural empirical bayes. Journal of Machine Learning Research, 20(181), 1-23.

---

> > ### Comment · Reviewer_bSDE · 2024-11-09
> >
> > I thank the authors for their comments and answers to my questions.
> >
> > I remain unconvinced about the significance of this work. The title of the work is "Gradient-guided discrete walk-jump sampling for **biological sequence generation**". Hence a reader would expect rigorous comparisons to existing works claiming to also do biological sequence generation. The admission that *"It is without a doubt that the metrics we benchmarked our results with has a little involvement in the protein design community"* is not encouraging for me to changing my rating of the audience standard.
> >
> > Furthermore, if you state that GFlowNets is performing de novo generation which is different than your task, then isn't it misleading to compare your approach to a method that isn't meant for your task?
> >
> > I disagree that the goal of GFlowNets and this work are fundamentally different. gg-dWJS is essentially the same as classifier guidance where the classifier is a reward function to which dd-dWJS is trying to optimize while producing diverse samples. This is the same goal as GFlowNets and other biological sequence generation methods.
> >
> > Lastly, you state "We choose the metrics for their fast evaluation following previous works (Stantonet al., 2022; Gruver et al., 2024),". I am well aware of these works. Neither use the secondary structure as a reward function. Stanton et al does optimize stability but for a different protein class, Red Fluorescent Proteins (RFPs). Furthermore, if the authors are proposing to follow prior works for evaluation then they should attempt to follow the evaluation procedures as closely as possible. As a result, I do not believe the claims are accurate as stated.

---

> ### Author Response · Authors · 2024-11-26
> **Q1 and Q2**
>
> Q1. I remain unconvinced about the significance of this work. The title of the work is "Gradient-guided discrete walk-jump sampling for biological sequence generation". Hence a reader would expect rigorous comparisons to existing works claiming to also do biological sequence generation.
>
> A1. We thank the reviewer for their insightful feedback. We believe "a reader would expect rigorous comparisons to existing works " is correct and we provide that through our baselines.
>
> Our method builds on discrete walk jump sampling paper for conditional generation, and indeed we provide the experiments comparing our method to the baseline, including a number of baselines of the previous work such as VDM, IgLM, and GPT 3.5.
>
> Again, we want to remind the reviewer their comment "Another issue is the baselines such as GPT-4o or VDMs which are not meant for biological problems" is conflicting with their recent comment "they should attempt to follow the evaluation procedures as closely as possible" as we follow the baselines provided by the work our method builds on.
>
>
> Q2. The admission that "It is without a doubt that the metrics we benchmarked our results with has a little involvement in the protein design community" is not encouraging for me to changing my rating of the audience standard.
>
> A2. We apologize for the misunderstanding. Our admission mainly resonates the limitation that our work does not provide any in-vitro experiments due to limited resources. This does not, however, mean that the contribution of our work is meaningless or not good enough. Indeed, the message that we want to bring to our community is that with some simple modification, discrete walk-jump sampling [1], can work for both single and multiple objective optimization and we provide examples of that as we quote reviewer E2Zk "Extensive experiments conducted on various datasets."
>
> Adding guidance to established methods is not rare, either. Indeed, [2] mentions in their opening,
>
> "Moreover, we use NOS to generalize LaMBO, a Bayesian optimization procedure for sequence design that facilitates multiple objectives and edit-based constraints."
>
> Besides, we do _not_ just utilize numerical objectives. We follow the established experiment performed by [1] for  generating CDR mutants on a hu4D5 antibody mutant dataset. This experiment requires us to train a model as a proxy for in silico experiments, where we show the single objective optimization capabilities of our method comparing it to many baselines reported in [1].
>
>
> Next, for our antibody tasks, we focus on attributes that are easily evaluable and choose attributes such as % beta sheets, instability index, molecular weight, grand average of hydropathy, and sequence length. Due to their easy to calculate nature, they have been used in previous experiments.
>
> Indeed, in [1], the authors write, "We considered sequence-based properties (calculated with BioPython (Cock et al., 2009)) of average hydprophilicity, molecular weight, grand average of hydropathy..."
>
> In [2], the authors convey, "we run experiments on two simple single-objective tasks...The percentage of beta sheets, measured on primary sequence"
>
> In [3], the authors mention, "to understand the biological relevance of the se- quences generated by GFlowNet-AL we study several physiochemical properties of the Top 100 generated sequences using BioPython. The instability index for the generated peptides is 26.5 on average with maximum of 36 (score of over 40 indicates instability). "
>
> In [4], the authors use proxy objectives too, saying, "We consider three objectives: the free energy of the secondary structure calculated with the software NUPACK, the number of base pairs and the inverse of the sequence length (to favour shorter sequences)."
>
> To sum up, our evaluations _do_ follow previous evaluations and _do_ utilize attributes that are followed by previous literatures. We thank the reviewer for raising this important question and are open to feedback, e.g., changing the paper title.
>
>
> 1. Frey, N. C., Berenberg, D., Zadorozhny, K., Kleinhenz, J., Lafrance-Vanasse, J., Hotzel, I., ... & Saremi, S. Protein Discovery with Discrete Walk-Jump Sampling. In The Twelfth International Conference on Learning Representations.
>
> 2. Gruver, Nate, et al. "Protein design with guided discrete diffusion." Advances in neural information processing systems 36 (2024).
>
> 3. Jain, Moksh, et al. "Biological sequence design with gflownets." International Conference on Machine Learning. PMLR, 2022.
>
> 4. Jain, Moksh, et al. "Multi-objective gflownets." International conference on machine learning. PMLR, 2023.

---

> ### Author Response · Authors · 2024-12-01
> **Q3 and Q4**
>
> Q3. Furthermore, if you state that GFlowNets is performing de novo generation which is different than your task, then isn't it misleading to compare your approach to a method that isn't meant for your task?
>
> I disagree that the goal of GFlowNets and this work are fundamentally different. gg-dWJS is essentially the same as classifier guidance where the classifier is a reward function to which dd-dWJS is trying to optimize while producing diverse samples. This is the same goal as GFlowNets and other biological sequence generation methods.
>
> A3. We do not think it is misleading to compare our approach to GFlowNets. This is indeed true that GFlowNets is discrete generative method and used for de novo generation tasks.
>
> We think that the confusion arises from the fact that our method uses score and discriminative models. If we consider the discriminative model as a reward function, the score model acts as a priori for previously seen samples (like in our case we have a large data of known antibodies). However, we agree that this problem can be bridged by having a differently chosen prior (like in our case we use initial sequence for our baseline).
>
> That said, we have carefully reviewed the reviewer's important suggestion and decided to perform experiments the reviewer has suggested. Particularly, we are going to perform the first experiment from the "Biological sequence design using GFowNets" paper that the reviewer has cited.
>
> To perform the experiment, we plan to utilize the same experimental setup that the paper mentions, namely having the D1 dataset as training samples and utilizing the D2 oracle for validation. In our case, we will use the D1 dataset for our score and discriminative model and test our results on the D2 oracle.
>
>
> We aim to update the results as soon as possible. We hope this will address the reviewer's concern and we apologize for the delay.
>
> Q4. Lastly, you state "We choose the metrics for their fast evaluation following previous works (Stantonet al., 2022; Gruver et al., 2024),". I am well aware of these works. Neither use the secondary structure as a reward function. Stanton et al does optimize stability but for a different protein class, Red Fluorescent Proteins (RFPs). Furthermore, if the authors are proposing to follow prior works for evaluation then they should attempt to follow the evaluation procedures as closely as possible. As a result, I do not believe the claims are accurate as stated.
>
> A4. Indeed, (Stantonet al., 2022) does not utilize the mentioned metrics. We apologize for the wrong citation. We will correct it with the correct citation which is [1].
>
> For instance, [1] mentions, "...to understand the biological relevance of the sequences generated by GFlowNet-AL we study several physiochemical properties of the Top 100 generated sequences using BioPython. The **instability index** for the generated peptides is 26.5 on average with maximum of 36 (score of over 40 indicates instability). "
>
> Besides, the claim that "Neither use the secondary structure as a reward function." is not true as well.
>
> (Gruver et al., 2024) mentions in their paper, "we run experiments on two simple single-objective tasks...**The percentage of beta sheets**, measured on primary sequence"
>
> We _do_, however, understand the confusion. We only utilize primary sequence for the objectives, so, indeed, we do not utilize the any secondary structures for the objectives. The calculated objectives serve as an easier alternative to perform experiments.
>
>
>
> We again thank the reviewer for their insightful and detailed observation. We request some time to perform the experiments mentioned above. We hope the reviewer finds our comments useful as the decision of whether our work gets accepted or not now depends on the reviewer. We hope to share our work with TMLR community.
>
> 1. Jain, Moksh, et al. "Biological sequence design with gflownets." International Conference on Machine Learning. PMLR, 2022.

---

> > ### Author Response · Authors · 2024-12-05
> > **AMP experiment results - submission update**
> >
> > We thank the reviewer again for suggesting us to perform discrete optimization tasks with established benchmarks such as the original GFlowNets paper. We updated the submission and you can now find the results in the section 7 and related details in the appendix C.3.
> >
> > We performed the AMP experiment detailed in GFlowNet-AL paper following the implementation as closely as possible. Table 7 illustrates our results measured over 10 random seeds. We see that our method significantly performs better than other baselines.
> >
> > We also provide physiochemical analysis as performed by the GFlowNet paper to understand the biological relevance of the sequences generated by gg-dWJS. We see that our method achieves better instability index compared to the baseline. Finally, in Figure 6, we show that our method's generated AMPs has a similar AA profile compared to that of known AMPs.
> >
> > Please let us know if you have any remaining doubts or concerns.

---

> ### Author Response · Authors · 2024-12-08
> **Reminder**
>
> Dear reviewer,
>
> We believe we have fulfilled all your concerns. Could you please let us know if there are any remaining concerns?

---

### Review · Reviewer_ZDHe · 2024-10-16

**Summary Of Contributions:**

This paper proposes gg-dWJS, a novel method for generating discrete sequences optimized for specific tasks. gg-dWJS extends the dWJS framework (Frey et al. 2024) by incorporating gradient information to guide the Markov Chain Monte Carlo (MCMC) sampling process on a smoothed data manifold. This guided sampling allows for generating high-quality discrete sequences, such as images and antibody sequences, tailored to particular objectives.

**Audience:**

Yes

**Broader Impact Concerns:**

No concerns.

**Claims And Evidence:**

Yes

**Requested Changes:**

**Requested Changes:**
1. Thorough revise and rewriting: The paper requires a major rewrite to improve clarity and accessibility. Focus on clearly defining all concepts, providing essential details, and ensuring a logical flow of ideas.
2. Detailed Explanation of dWJS: Provide a comprehensive introduction to dWJS, including all necessary components and the mechanism for transitioning between continuous to discrete spaces.
3. Clarify Connection to *Classifier Guidance*: Explicitly discuss the relationship between gg-dWJS and classifier guidance, highlighting similarities and differences.
4. Precise Notation: Ensure all notation is clearly defined and consistently used throughout the paper.
5. Elaborate on Experimental Setup: Provide more details on the experimental setup, including the specific objectives, and how the guidance where trained and evaluated (e.g., how good shall this classifier needs to be?)
6. Expand Discussion: Deepen the discussion on the trade-off between data fidelity, diversity, and optimization, and provide insights into the limitations and future directions of the work.

Other Comments:
- Section 2.1: Define the meaning of the "half arrow."
- Section 2.3: Provide a precise definition of the max operation over the vector R(x). This seems not well defined.
- Section 3: Clearly define P(C|Y), c, C, y, and Y.  Clarify the sentence *"Formally, we use (Math expression) to perform MCMC for K steps…"* (It is not clear what does this mean in practice)?
- Algorithm 1: Explain the rationale behind initializing y_0 as the sum of noise and a uniform random variable.
- Section 5: Clarify the term "denoising walk" and its connection to denoising in Neural Empirical Bayes (Tweedie's formula). Provide more details on the objective used in the antibody sequence generation task (how do you train the classifier models?).
- Section 6.3: Clearly define the role of w in the multi-objective optimization.
- Section 6.4: Suggested change "Effect of \lambda" to "Effect of the guidance strength."
- Section 8: Expand the discussion on the trade-off between diversity and optimization.

**Strengths And Weaknesses:**

**Strengths**:
- The paper introduces an interesting idea by combining gradient guidance with dWJS for discrete sequence generation.
- Diverse Applications: The method is evaluated on a range of tasks, including image and antibody sequence generation, demonstrating its versatility.
- Multi-objective Optimization: The paper explores the application of gg-dWJS in multi-objective settings, an important and challenging area.

**Weaknesses:**
- Presentation and Clarity: The major weakness is the paper's presentation. Several components are not correctly defined, important details are missing, and the overall exposition needs significant improvement. This severely hinders the understanding and appreciation of the contributions.
- Lack of Connection to Related Work: The paper lacks a clear connection to related work, particularly *classifier guidance*. The relationship between the proposed method and existing techniques should be explicitly discussed.
- Insufficient Explanation of dWJS: As gg-dWJS builds upon dWJS, a thorough introduction to the latter is crucial. The paper only partially introduces dWJS in Section 2.2, leaving out important details about the transition from continuous to discrete representations.
- Ambiguous Notation: See comments below.
- Lack of Clarity. For instance:
  - The term "denoising walk" in Section 5 needs clarification.
  - The specific objective used in the antibody sequence generation task requires more detail ("For each task, we train a smoothed predictor on the antibody sequences and use its gradient guidance to optimize the single objective".)
  - The role of w in the multi-objective optimization (Section 6.3) is unclear. Does  w represent the probability of choosing one objective or the other?

---

> ### Author Response · Authors · 2024-10-21
> **Reply to the reviewer ZDHe**
>
> Dear reviewer HDHe,
>
> The insights you provided in the reviews are invaluable, and we would like to thank you for your time crafting this review. Please go over the changes we made in blue.
>
> In what follows, we address your comments one by one.
>
> ----------
>
> **RC1.  Thorough revise and rewriting**
>
> We have performed a through revise as suggested. Additionally we also went through the suggested comments and fixed these.
>
> ----------
>
> **RC2. Detailed Explanation of dWJS:**
>
> Thank you for your suggestion. We have added a more detailed explanation of WJS, dWJS and their connection to our work in **Section 2**.
>
> ----------
>
> **RC3.  Clarify Connection to  _Classifier Guidance_**
>
> We agree that the connection to classifier guidance is important as our work is motivated by it. As suggested, we have added a new subsection in **Section 2** to discuss it.
>
> ----------
>
> **RC4. Precise Notation**
>
> Thank you for suggestion and pointing out the confusion in the weakness section. We reviewed the weakness and revised our work accordingly. **Section 3** first paragraph contains a clear description of the notations. We have also added a table of all notations in the **Table 7** in the **Appendix A**.
>
> ----------
>
>
> **RC5.  Elaborate on Experimental Setup**
>
> Thank you for your suggestion! We provide more details on the training in the **Appendix C1** and **Appendix D**. As suggested, we mainly touch on the training details such as batch size and number of epochs, as well as training objectives such as the different training losses.
>
> _______
>
> **RC6. Expand Discussion:**
>
> Thank you. It helps to expand the future works. As suggested, we have gone deeper into the future works involving fidelity, diversity, and optimization in the **Section 8**.
> _______
>
>
> **Other comments**
>
> - *Section 2.1: Define the meaning of the "half arrow."*
>
> We have now added the definition in **Section 2.1**.
> - *Section 2.3: Provide a precise definition of the max operation over the vector R(x). This seems not well defined.*
>
> We have now added the definition in **Section 2.4**.
> - *Section 3: Clearly define P(C|Y), c, C, y, and Y. Clarify the sentence  _"Formally, we use (Math expression) to perform MCMC for K steps…"_  (It is not clear what does this mean in practice)?*
>
> We have clarified it in **Section 3**.
> -  *Algorithm 1: Explain the rationale behind initializing y_0 as the sum of noise and a uniform random variable.*
>
> In the discrete space, an unform random variable puts similar probability to all the possible options. Since y is the smoothed sample from the true data, we add gaussian noise to smoothen it.
> -  *Section 5: Clarify the term "denoising walk" and its connection to denoising in Neural Empirical Bayes (Tweedie's formula). Provide more details on the objective used in the antibody sequence generation task (how do you train the classifier models?).*
>
> We have added more details in the **Section 5**, **Appendix C, and Appendix D**.
> -  *Section 6.3: Clearly define the role of w in the multi-objective optimization.*
>
> We have added more details in **Section 6.3**.
>
> -  *Section 6.4: Suggested change "Effect of \lambda" to "Effect of the guidance strength."*
>
> Thank you. We have changed the Section title accordingly.
> -  *Section 8: Expand the discussion on the trade-off between diversity and optimization.*
>
> We have added more details in **Section 8**.

---

### Review · Reviewer_E2Zk · 2024-11-04

**Summary Of Contributions:**

This paper build upon existing work dWJS to improve its capacity for conditional generation. Experiments on various discrete data forms have shown its good empirical performances.

**Audience:**

Yes

**Claims And Evidence:**

No

**Requested Changes:**

See above.

**Strengths And Weaknesses:**

**Strengths**:
1. Well-written and easy to follow.
2. Extensive experiments conducted on various datasets.

**Weaknesses**:
1. What is the core difference from classifier-guided generation? Steps A-C appear to be standard design principles for diffusion models.

2. Explain the specific steps used in the "Gradient-guided jump" more clearly.

3. The transition back to discrete data has been explored in previous work using continuous-state models to represent discrete data as well [1-2]. It would be beneficial to cite these works and include a brief discussion.

4. Compared to dWJS, is the proposed method an adaptation for conditional generation? What is its most significant novelty?

**References**:
[1] Yan, Q., Liang, Z., Song, Y., Liao, R., and Wang, L. SwingNN: Rethinking permutation invariance in diffusion models for graph generation. TMLR.

[2] Niu, C., Song, Y., Song, J., Zhao, S., Grover, A., and Ermon, S., 2020, June. Permutation Invariant Graph Generation via Score-Based Generative Modeling. In *International Conference on Artificial Intelligence and Statistics* (pp. 4474-4484). PMLR.

---

> ### Author Response · Authors · 2024-11-10
>
> Dear reviewer E2Zk,
>
> Thank you for your encouraging comments! The insights you provided in the reviews are invaluable, and we would like to thank you for your time crafting this review. Please go over the changes we made in blue.
>
> In what follows, we address your comments one by one.
>
> ----------
>
> **RC1. Core difference between classifier guidance and gg-dWJS**
>
> We agree that our gradient guidance takes inspiration from the classifier guidance. However, unlike denoising diffusion, dWJS has two steps that enable it to be applicable on the discrete data at any given point in time. In gg-dWJS, thus, gradient guidance is applied not only during the sampling (walking) but also in the jump back to the discrete data manifold. Specifically, gg-dWJS leverages both a score model and a discriminator model for gradient guidance, combining these to jump back to the discrete manifold at any point in time, unlike classifier guidance in diffusion.
>
>
> ----------
>
> **RC2. Clarify gradient guided jump**
>
> Thank you for your suggestion to make our work clearer. We have added more details about the gradient guided jump, especially why it is required in our context, in the **Section 3.4**.
>
>
> ----------
>
> **RC3. Discussing previous works that transition between continuous to discrete**
>
> Thank you for your suggestion. The works you mentioned indeed tackle a similar problem. We have added a brief discussion of the works (as well as cite them) in the **Section 7.1**.
>
> ----------
>
> **RC4. Explaining novelty compared to dJWS**
>
> Our method is an adaptation for conditional generation for dWJS. While similar, it is not straightforward (e.g., the formulation of walk and jump) to adapt classifier guidance to dWJS. Besides, we also adapt our method for preference conditioning, which is useful for multi-objective optimization.
>
> Please let us know if there is anything left unclear.

---

> > ### Comment · Reviewer_E2Zk · 2024-11-10
> >
> > Thank you for your response.
> >
> > In Section 3.4, how can we ensure that the point $\hat{x}$ lies on the desired discrete manifold? The right-hand side of Equation (4) consists of continuous values or neural functions, which can potentially yield arbitrary values. Technically, is any form of thresholding or enforced discretization needed to return to the discrete data manifold?
> >
> > I also have a follow-up question on preference conditioning: while including a reward objective for controllable generation in generative models is appealing, in the experiments presented in Section 6.3, the weight $w$ was chosen as $w \in \{[0, 1], [1, 0], [0.5, 0.5]\}$. Due to the sum-to-one constraint, this essentially reduces to adding a single scalar input to the model as conditioning. Could you provide larger-scale experiments to support the objectives and claims in Sections 2.4 and 4 on multi-task optimization?

---

> > > ### Author Response · Authors · 2024-11-11
> > >
> > > Thank you for your questions!
> > >
> > > > In Section 3.4, how can we ensure that the point lies on the desired discrete manifold? The right-hand side of Equation (4) consists of continuous values or neural functions, which can potentially yield arbitrary values. Technically, is any form of thresholding or enforced discretization needed to return to the discrete data manifold?
> > >
> > > You are correct to point out that we are operating on the continuous space, even in the gradient-guided jump step which should lead to the discrete space. To address this, we simply take the $\texttt{argmax}$ of the categorical values to go back to the discrete space after a gradient-guided jump, following previous work [1]. *Whether* the gradient-guided jump step will bring us to the discrete space can be answered by [2], which shows that the least square estimator (in this case, the learnt score model) can bring us to the discrete manifold at an arbitrary time.
> > >
> > > > I also have a follow-up question on preference conditioning: while including a reward objective for controllable generation in generative models is appealing, in the experiments presented in Section 6.3, the weight $w$ was chosen as $w \in \\{[0, 1], [1, 0], [0.5, 0.5]\\}$. Due to the sum-to-one constraint, this essentially reduces to adding a single scalar input to the model as conditioning. Could you provide larger-scale experiments to support the objectives and claims in Sections 2.4 and 4 on multi-task optimization?
> > >
> > > Thank you for the question. Could you please help us understand your question better? For example, the sum-to-one constraint you mentioned *is* an assumption that we made in **Section 2.4** as part of the preference conditioning and is found in previous works too [3]. Would you like to see $w$ to be more diverse than our chosen three?
> > >
> > >
> > > Refereneces
> > >
> > > 1. Frey, N. C., Berenberg, D., Zadorozhny, K., Kleinhenz, J., Lafrance-Vanasse, J., Hotzel, I., ... & Saremi, S. Protein Discovery with Discrete Walk-Jump Sampling. In The Twelfth International Conference on Learning Representations.
> > > 2. Saremi, S., & Hyvärinen, A. (2019). Neural empirical bayes. Journal of Machine Learning Research, 20(181), 1-23.
> > > 3. Jain, Moksh, et al. "Multi-objective gflownets." International conference on machine learning. PMLR, 2023.

---

> > > > ### Comment · Reviewer_E2Zk · 2024-11-15
> > > >
> > > > Thanks for your response.
> > > >
> > > > 1. Please consider including the argmax operation directly in the main text for better algorithm clarity.
> > > >
> > > > 2. As I understand it, $w$ can generally be an $n$-dimensional vector for $n$ tasks. In the presented experiments, $n$ is set to 2. I was wondering if it would be possible to include additional experiments with higher values of $n$ to observe how this affects performance.

---

> > > > > ### Author Response · Authors · 2024-11-22
> > > > >
> > > > > Thank you for suggestions!
> > > > > > Please consider including the argmax operation directly in the main text for better algorithm clarity.
> > > > >
> > > > > We have now added it. Our current draft should reflect your suggestion.
> > > > >
> > > > > > As I understand it,  can generally be an -dimensional vector for  tasks. In the presented experiments,  is set to 2. I was wondering if it would be possible to include additional experiments with higher values of  to observe how this affects performance.
> > > > >
> > > > > We have now performed additional experiments and added the experimental results in the **Table 12** of the **Section E3**. For experiment, we chose five objectives: target molecular weight, inverse sequence length, negative GRAVY, percentage beta sheets, and inverse instability index. So, n = 5 for our experiments. We hope our experiment addresses your concerns.

---

> > > > > > ### Comment · Reviewer_E2Zk · 2024-11-25
> > > > > >
> > > > > > Thanks for your updates.

---

> > > > > > > ### Author Response · Authors · 2024-11-25
> > > > > > >
> > > > > > > Thank you for responding. Do our efforts address your concerns?

---

> > > > > > > > ### Comment · Reviewer_E2Zk · 2024-11-25
> > > > > > > >
> > > > > > > > Yes, your comments have addressed my concerns. However, I am also interested in Reviewer bSDE's feedback on the manuscript. It would be great if you could address Reviewer bSDE's queries as well to further improve the paper.

---

> > > > > > > > > ### Author Response · Authors · 2024-11-25
> > > > > > > > >
> > > > > > > > > Thank you. We are working on it.

---

> ### Author Response · Authors · 2024-12-05
>
> We believe we have addressed all of reviewer bSDE's queries, including performing one more experiment.
>
> We hope that makes our claim stronger and encourages you to update the rating of our paper. Thank you again for your patient feedback and careful watch on our paper!

---

### Comment · Reviewer_E2Zk · 2024-10-22

One quick question: why is there so much blue text in the PDF? Could the AE please remind the author to ensure they are submitting the correct version of the manuscript?

---

> ### Author Response · Authors · 2024-10-22
>
> Dear Reviewer E2Zk,
>
> Based on the previous reviews and suggestions, we have updated our manuscript accordingly. We mark the changes with blue text. Please let us know if that is inconvenient.

---

> > ### Comment · Reviewer_E2Zk · 2024-10-22
> >
> > Got it. Thanks for the clarification.

---

### Author Response · Authors · 2024-12-07
**Summary of rebuttal**

Dear Action Editor,

We appreciate the reviewers' thoughtful evaluations of our paper and have carefully addressed their concerns. Below, we provide a unified discussion of the main issues raised and our responses.
______

Main Concerns and Our Responses:

- Clarification of Contributions and Claims:
  - Concern: Some reviewers noted that our method is similar to classifier guidance in diffusion.
  - Response: We agreed that our gradient guidance takes inspiration from the classifier guidance. However, unlike denoising diffusion, dWJS has two steps that enable it to be applicable on the discrete data at any given point in time. In gg-dWJS, thus, gradient guidance is applied not only during the sampling (walking) but also in the jump back to the discrete data manifold. Specifically, gg-dWJS leverages both a score model and a discriminator model for gradient guidance, combining these to jump back to the discrete manifold at any point in time, unlike classifier guidance in diffusion. So, while similar, it is not straightforward (e.g., the formulation of walk and jump) to adapt classifier guidance to dWJS. Besides, we also adapt our method for preference conditioning, which is useful for multi-objective optimization.

- Scale of multi-objective optimization experiments:
  - Concern: Reviewer E2Zk asked for larger-scale experiments to support the objectives and claims in Sections 2.4 and 4.
  - Response: We expanded our experiments by performing additional experiments with higher number of objectives: : target molecular weight, inverse sequence length, negative GRAVY, percentage beta sheets, and inverse instability index. We showed that our method is able to sample distinct sequences based on the weight of objectives.

- Significance of improvements:
  - Concern: Questions were raised about the improvements achieved by our method in objectives such as instability index and beta sheet percentage.
  - Response: We incorporated paired t-test results with our results in the single objective experiments, showing that the improvements achieved by our method is statically significant.

- Relevance to the protein engineering community:
  - Concern: Reviewer bSDE felt that our method is limited in its significance to the protein engineering community.
  - Response: We acknowledged that the message that we want to bring to our community is that with some simple modification, discrete walk-jump sampling can work for both single and multiple objective optimization.
     Besides, we clarified that except for the multi-objective experiments, all the experiments performed in our paper follows previous literatures. We also showed with reference that the attributes we chose are indeed utilized in previous literatures.
Adding to this, we performed experiments suggested by reviewer bSDE and showed that our method performs better than other methods in an established benchmark.

- Clarity and Presentation Enhancements:
  - Concern: Suggestions were made to improve training descriptions, algorithm steps, and training details.
  - Response: We revised Section 2 to reflect preliminary works and our motivations better, including adding classifier guidance. Besides, we updated Section 3 to improve the readability of our method descriptions. We changed Algorithm 1 to reflect that we use Argmax. We also revised Section 9 and 10 to expand upon the related works and future works.

We believe that our revisions have effectively addressed the reviewers' concerns and have strengthened the paper's clarity and contribution to the field.

We kindly ask the action editor to consider our responses and the improvements made.

Thank you for your consideration!

---

### Decision · Action_Editor_tLKy · 2024-12-16

**Recommendation:** Accept with minor revision

**Comment:**

There was a significant discussion of the paper among the reviewers and I commend the authors and reviewers for thoroughly evaluating the paper. The main concerns were around the degree of technical novelty and scale of the experiments. The authors have addressed the latter with a significantly expanded set of experiments. The former seems to a matter of degree. There was some consensus that the novelty may not be high. However, since this work is highly interdisciplinary, novelty is also a matter of the perspective of the reviewer. Some may view the approach as incremental, while others may view the approach as more novel in the peptide generation area. I recommend acceptance of the revised manuscript taking into account the evaluation policy of the journal and the challenges of judging novelty of interdisciplinary work.

**Audience:**

The audience is researchers interested in using machine learning methods for protein and peptide engineering. There is sufficient interest in the topic among the audience of the journal.

**Claims And Evidence:**

The reviewers offered significant recommendations to address the comments and engaged in a dialogue with the authors.

The main claims of the paper are:
1. a novel algorithm based on dWJS for target optimization using a noised discriminator model.
2. a representation of the method for use in a multi-objective optimization setting.
3. an evaluation of the method on discretized image, antibody sequence generation, and peptide generation tasks
4. the method generates high-quality samples with optimized attributes

The first claim is in regard to the novelty of the method. Several reviewers raised concerns about novelty and some comments were around significance. The journal policy on evaluation is that (1) the claims are well-supported and (2) some of the journal audience would be interested. The policy specifically states, "Papers should be accepted if they meet the criteria, even if the contribution or significance of the work is modest." In my assessment, the authors have clearly stated what the novelty of their method is. As this is a paper at the intersection of computational biology and machine learning, there may be some in the community who view the improvements as more novel than others and perhaps more significant in terms of utility. Therefore, there seems to be sufficient evidence to support the claim of novelty and the degree of novelty and significance is best judged by each reader.

The second and third claims are well-supported by the evidence in the paper. The authors have conducted additional experiments to support their claims during the review period.

The final claim is more subjective and evidenced by Figure 3 and other data. It is expected that the quality of the generated peptides would best be assessed by experimental binding studies or other biological validation that would be outside of the scope of this work. If resources and collaborations are available, the authors are encouraged to pursue such studies to fully validate their approach. However, in the context of this work, the claim of quality and attributes seem aligned with the evidence.